# A Scoping Review on COVID-19-Induced Cardiovascular Complications

Ian Osoro [1], Manisha Vohra [1], Mohammad Amir [1], Puneet Kumar [2] and Amit Sharma [1,*]

1 Department of Pharmacy Practice, ISF College of Pharmacy, Moga 142001, Punjab, India
2 Department of Pharmacology, Central University of Punjab, Ghudda 151401, Punjab, India
* Correspondence: choice.amit@gmail.com; Tel.: +91-9418783145

**Abstract:** Severe acute respiratory syndrome coronavirus 2 (SARS-CoV-2) is a type of human coronavirus that resulted in the 2019 coronavirus disease (COVID-19). Although it was generally categorized as a respiratory disease, its involvement in cardiovascular complications was identified from the onset. Elevated cardiac troponin levels (a myocardial injury marker) and echocardiograms, which showed the anomalous performance of the patients' hearts, were noted in the early case reports obtained from Wuhan, China. A couple of mechanisms have been proposed to explain COVID-19-induced cardiovascular complications, with systemic inflammation being the major focus recently. Chest pain and palpitations are among the prevalent symptoms in moderate to severe COVID-19-recovering patients. Cardiac damage potentially occurs due to multifactorial factors, which include cytokine-induced inflammation, direct cardiotoxicity, and disseminated intravascular coagulation (DIC), among others. The cardiovascular manifestations include cardiac arrhythmia, cardiogenic shock, venous thromboembolism, and elevated cardiac biomarkers. Both the long- and short-term effects of these cardiovascular complications remain puzzling to researchers, as substantial evidence is yet to be gathered to reach a consensus on the severity of COVID-19 in the heart. The treatment considerations currently include antiarrhythmic management, ACEI or ARB use, anticoagulation, hemodynamic support, and immunosuppression. This review aimed to outline the pathogenesis of the various cardiac complications due to COVID-19 as well as the available treatment modalities of COVID-19 infection. Both the mechanisms and the treatments have been succinctly explained in a proper manner to ensure understanding.

**Keywords:** SARS-CoV-2; cardiovascular complications; ACE-2 receptors; myocardial injury; COVID-19; cardiomyocytes; improved patient outcomes

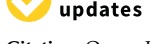



## 1. Introduction

As of 22 August 2022, Turkmenistan remains the only country without any COVID-19 reported cases, according to WHO reports. This is exceptional, especially because it is already about 2.5 years since the COVID-19 pandemic was declared, and currently, there is a total of about 591 million confirmed cases globally, with the United States of America leading, recording the highest number of cases. COVID-19 is caused by severe acute respiratory syndrome coronavirus 2 (SARS-CoV-2), and it was first reported to the WHO as pneumonia of unknown cause on 31 December 2019 from Wuhan city, China [1]. SARS-CoV-2 is a single-stranded and RNA-enveloped virus [2]. Its resultant infection includes symptoms such as fatigue, dry cough, fever, and anorexia [3,4]. The family of SARS-CoV viruses has been identified to infect humans earlier on, and SARS-CoV-2 is the seventh known human coronavirus [5].

Cardiovascular complications in COVID-19 patients have been common, and they are among the highest comorbidities in COVID-19 patients. In a study conducted by the Chinese Centre for Disease Control and Prevention, about 44,672 laboratory-confirmed

cases were evaluated, and 4.7% of cases were classified as critically ill (multiorgan failure, septic shock, and/or respiratory failure), having the potential of advancing to acute respiratory distress syndrome (ARDS). Higher mortality and more severe cardiovascular complications were observed in COVID-19 patients with cardiovascular comorbidities [6,7]. Moreover, the development of cardiovascular complications during COVID-19 was noted in patients without a CVD history, which could ultimately result in detrimental outcomes during the infection. Cardiac biomarkers elevation (which is used to predict mortality), as well as vascular injury, may be caused by COVID-19, hence the need to understand the mechanisms behind these clinical manifestations [8,9]. In the most critically ill patients, CVD was found to be among the major cause of death in COVID-19 patients [10–12].

## 2. Materials and Methods

A rapid evidence review was conducted, and the scientific literature used mainly constituted original articles sourced from online databases like (PubMed, Cochrane, Web of Sciences, and Google Scholar). Then a careful screening based on *the Preferred Reporting Items for Systematic Review and Meta-Analysis Protocols (PRISMA-P)* was conducted so as to compound the wide pool of information collected, as shown in Figure 1 below. Keywords such as "COVID-19, Cardiovascular complications, recent therapy in COVID-19 treatment, mechanisms of COVID-19" were used.

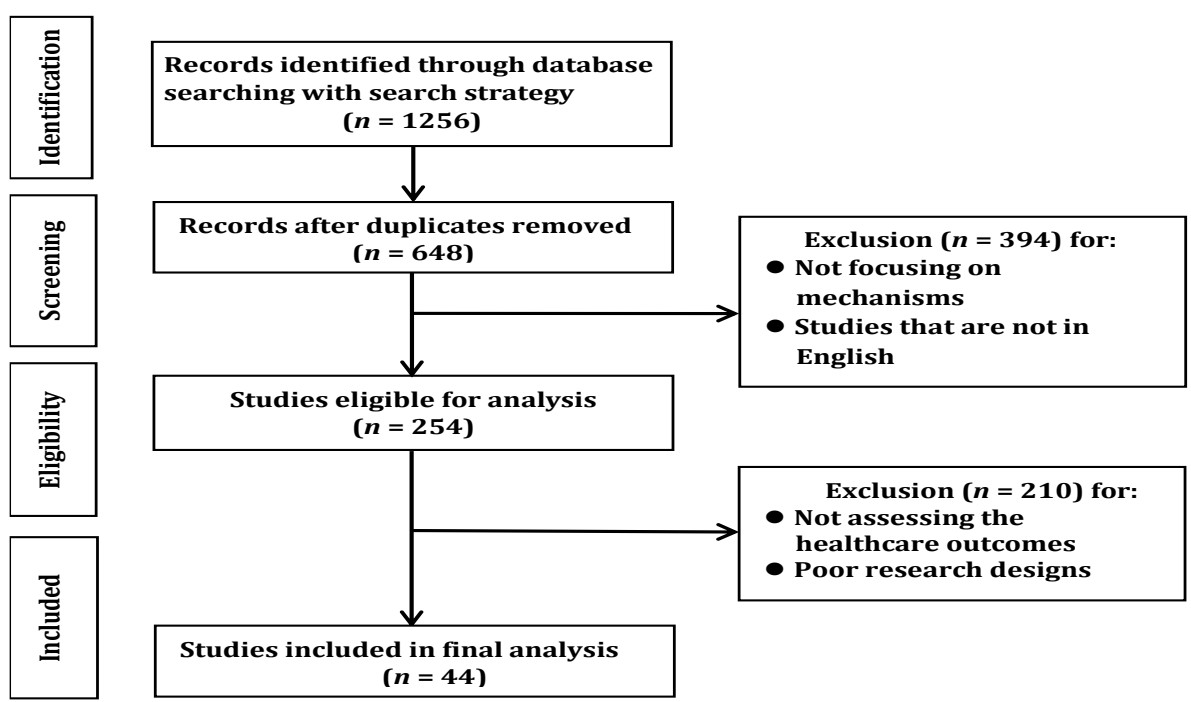

**Figure 1.** Preferred Reporting Items for Systematic Review and Meta-Analysis Protocols (PRISMA-P).

## 3. Mechanisms of Cardiovascular Complications in COVID-19

### 3.1. ACE-2 Inhibitors Downregulation

There are still divergent perspectives on the COVID-19 pathophysiology; however, prior epidemics (SARS and MERS) have been of great benefit in the comprehension of the SARS-CoV-2 receptor recognition and its mechanism [13]. ACE-2 is a bioactive peptide that participates in the progression of cardiovascular diseases through the RAS system and effectually binds with SARS-CoV S protein [14,15]. The transmembrane spike glycoprotein has its domain (S1) exposed on the viral surface, and with it being enzymatically active, it mediates viral entry through the receptor-binding domain [16]. ACE-2 has been proven to be a specific functional receptor for SARS-CoV [14,17]. SARS-CoV-2 entry in cells without ACE-2 receptors, such as dipeptidyl peptidase 4 (DPP4), is impossible, whereas it easily

enters ACE-2 receptor-expressing cells, which validates ACE-2 as the cell receptor for SARS-CoV-2 [18]. The fusion of the host cell and virus membrane causes the release of the viral RNA to the cytoplasm, resulting in infection. In SARS-CoV infection, ACE-2 is internalized along with the virus; however, its peptidase activity and catalytically active site are not distorted due to the binding process. Transmembrane proteinases (such as ADAM17 and TMPRSS2) and proteins (such as clathrin) are possibly involved in the binding and membrane fusion process [19–21]. A disintegrin and metallopeptidase domain 17 [ADAM17] leads to ectodomain shedding by cleaving to ACE-2, whereas transmembrane protease serine 2 [TMPRSS2] leads to viral uptake enhancement when it cleaves to ACE-2 [18,22]. Single-cell RNA-seq analysis and immuno-histochemical analysis application in various human cells disclose that the alveolar epithelial cells produce higher expression of ACE-2 than in oral and nasal mucosa epithelial cells, which blatantly points out the fact that the main target of SARS-CoV-2 is the lungs [23]. The heart may also be the first-hand target of SARS-CoV-2 because the cardiomyocytes, endothelial cells, and pericytes that are located in the heart express ACE-2 [15,24]. This has been well depicted, as per Figure 2 below.

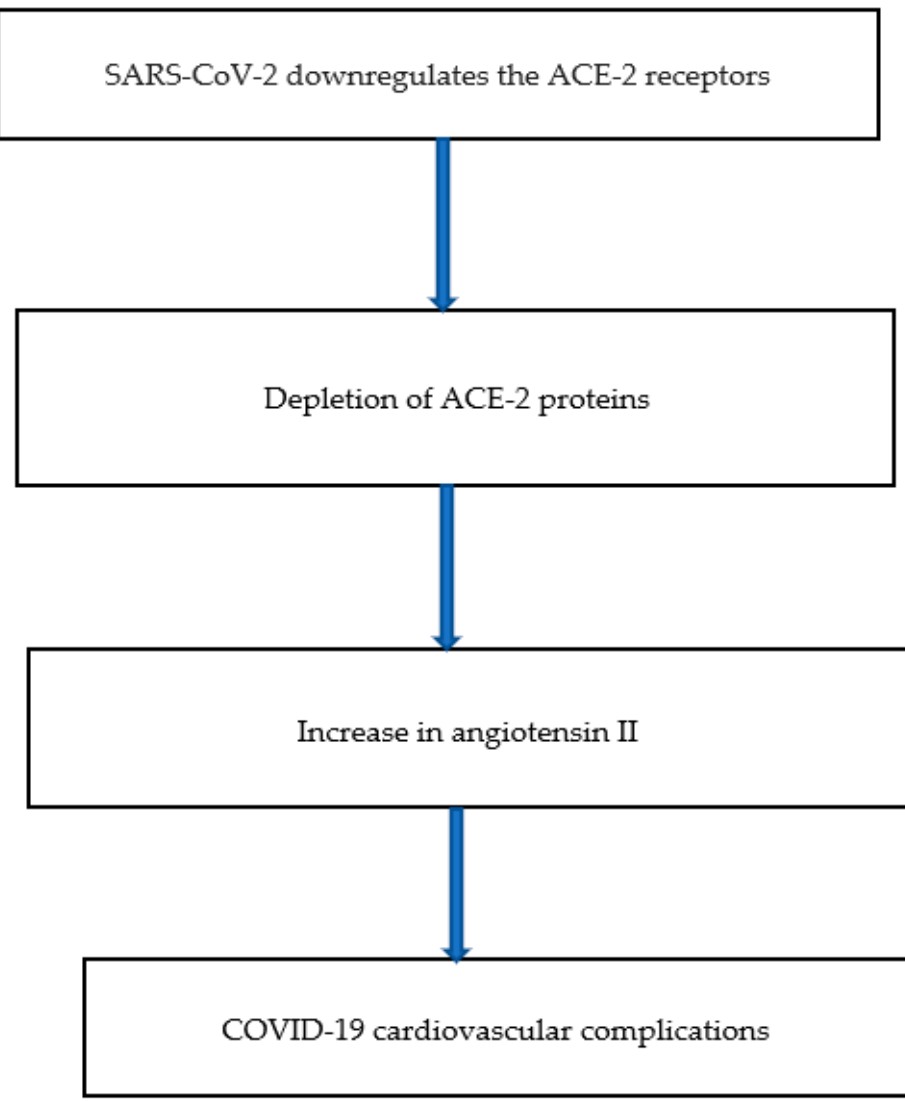

**Figure 2.** Diagrammatic illustration of ACE-2 inhibitors downregulation mechanism.

SARS-CoV-2 downregulates the ACE-2 receptors. ACE-2 is significant in the degradation of angiotensin II, resulting in heptapeptide production, which is referred to as angiotensin 1–7. Therefore, an imbalance between ACE-2 and angiotensin II levels is noted. This imbalance is responsible for the cardiovascular complications in COVID-19 patients, including those with no prior CVD history and the aggravated condition in those with existing CVD. The anti-inflammatory and vasodilator effects on the cardiovascular system are observed when this heptapeptide binds to G protein-coupled receptor [25,26].

Oudit et al. demonstrated that SARS-CoV could affect the heart, resulting in ACE-2 protein level depletion [27]. Interestingly, autopsy reports from the hearts of SARS-CoV-2 infected patients showed notably depleted ACE-2 levels.

### 3.2. Cytokine Storm

Cytokine release syndrome (CRS) is an inflammatory response that is systemic, and it was first explained during the use of anti-T-cell antibody muromonab-CD3 (OKT3), a drug used after an organ transplantation has taken place [28]. CRS is meant to be a form of an inherent immune response, but it can become fatal when there are complications in various viral infections. Multiple organ failure and ARDS (acute respiratory distress syndrome) are some of the undesired effects of CRS. It is important to note as well that CRS was observed in both MERS and SARS epidemics as it is also being reported in SARS-CoV-2 [29]. CRS can either be mild, wherein the clinical manifestations include fatigue, rash, and arthralgia, or it can be severe, wherein the clinical manifestations will include organ failure, DIC (disseminated intravascular coagulopathy), vascular damage, and even shock [30].

CRS may cause cardiovascular complications by promoting cardiac toxicity, and the resultant cardiomyopathy resembles Takotsubo cardiomyopathy. Tachycardia, arrhythmia, and reduced ejection fraction may be experienced during this cardiac malfunction. The laboratory tests normally depict elevated C-reactive protein levels, distorted D-dimer values, and altered prothrombin time [29,31]. Laboratory investigations showed that lymphocytopenia in COVID-19 patients is linked with the severity of the symptoms manifested [32]. Contrary to expectation, the CRS fails to protect the body but destroys it instead through an exceeded immune response. Low cytokine and chemokine levels are initially produced after the COVID-19 infection in the respiratory epithelial cells and macrophages. Elevated proinflammatory cytokine levels (IL-1β, IL-6, and TNF) with reduced antiviral factors interferons (IFNs) levels are subsequently observed as the mechanism progresses. Inflammatory cells (neutrophils, monocytes) are enticed with high levels of chemokines and cytokines, causing lung tissue infiltration and, ultimately, lung injury. This macrophage and neutrophil intrusion into the lungs causes diffuse alveolar destruction, which culminates in ARDS. Studies have depicted that IL-6 levels correspond emphatically to disease severity, and drugs that inhibit IL-6 secretion should be considered in CRS treatment [33]. This has been well depicted, as per Figure 3 below.

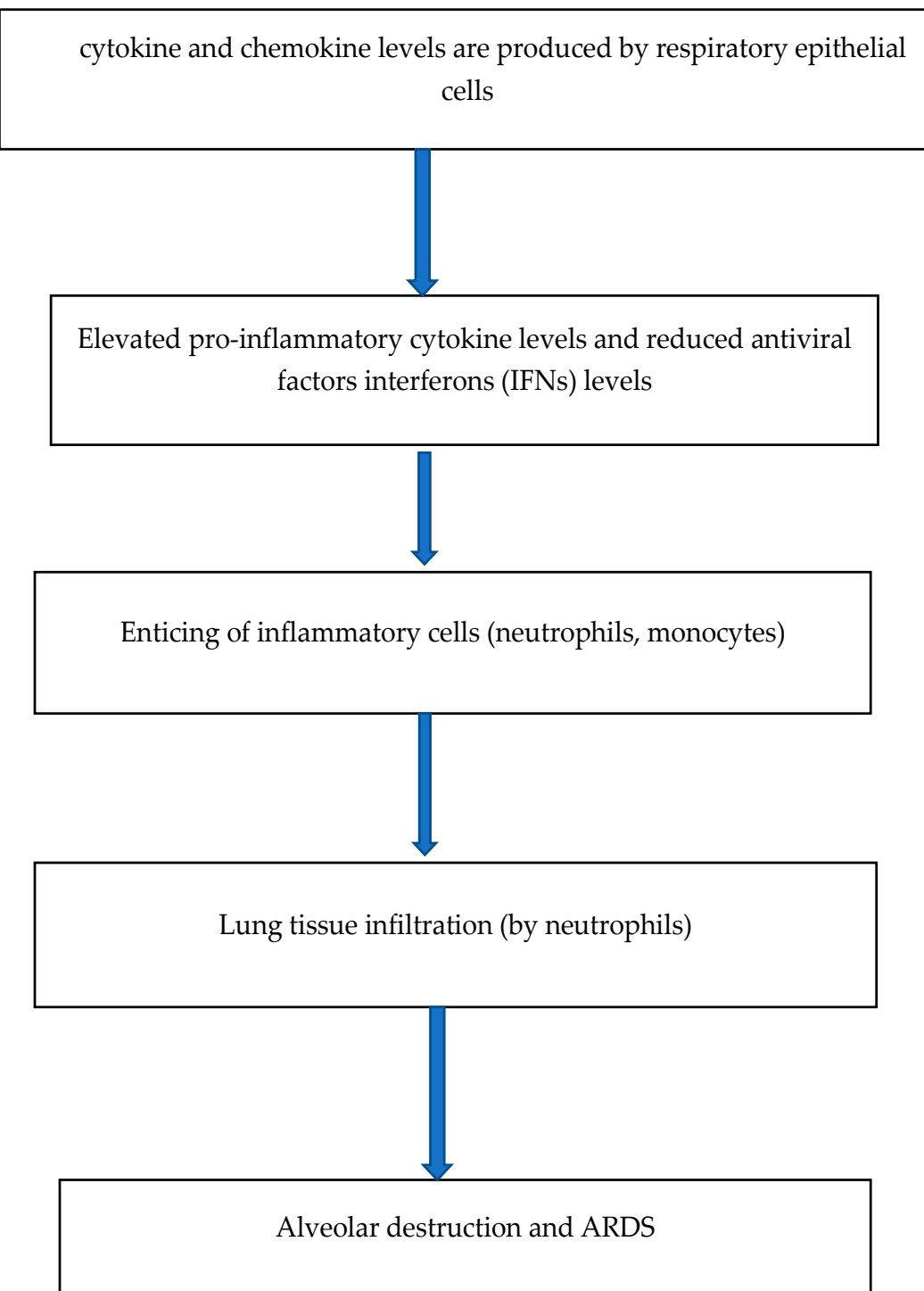

**Figure 3.** Diagrammatic illustration of cytokine storm mechanism.

*3.3. Plaque Alteration*

The elevated catecholamine levels due to systemic inflammation (as depicted in COVID-19 patients) may cause acute coronary syndrome by rapturing the plaque [34]. The C-reactive protein levels have a direct connection with the onset of myocardial infarction as a result of plaque rapture [35]. According to a study by Wang et al., elevated C-reactive protein in COVID-19 patients influences the severity of the clinical manifestation [36]. When the plaque ruptures, there is exposure to macrophages that are based under the endothelium. Microthrombi development begins due to the blood coming into contact with the expressed tissue factor [37]. Moreover, an acute response may be activated due to IL-6

over-production if the smooth muscle cells undergo inflammatory activation [38]. Figure 4 below shows this mechanism.

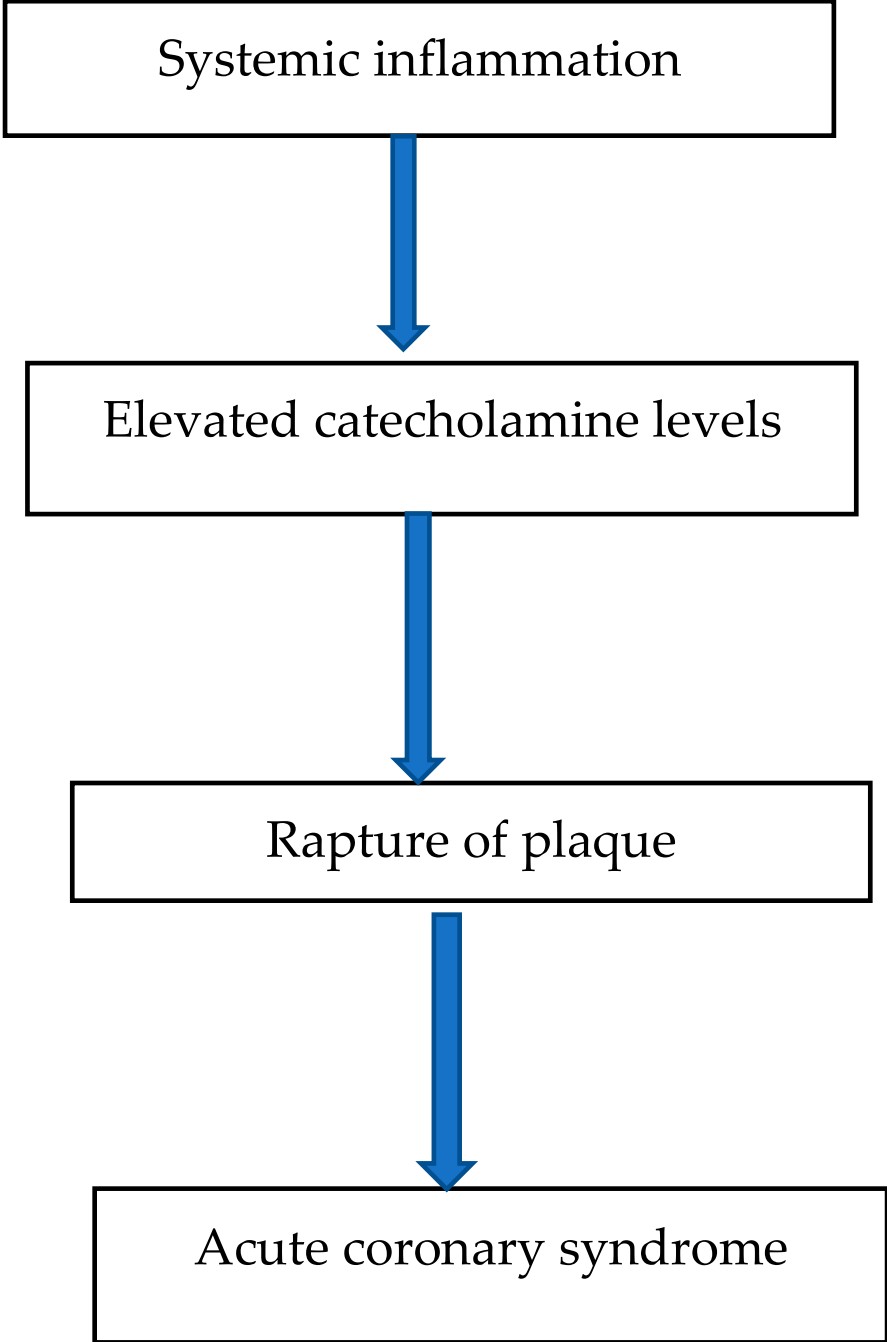

**Figure 4.** Diagrammatic illustration of plaque alteration mechanism.

*3.4. Prothrombotic State and Hypercoagulability*

Microthrombi development occurs as a result of systemic inflammation, and it may lead to an impediment to the normal functioning of various organs. The hematopoietic system also undergoes restriction due to this inflammation, which is normally caused by infections [39].

Significantly elevated D-dimer levels (36%) and elevated incomplete thromboplastin time (6% of patients) were seen in a study involving 99 COVID-19-infected patients in Wuhan [40]. Moreover, a meta-analysis study showed that there was a strong correlation between D-dimer levels and the severity of COVID-19 in patients [41]. Another study was

conducted to measure the coagulation parameters in 216 COVID-19-confirmed cases, and the results showed that 44 patients (20%) experienced prolonged aPTT time (activated partial thromboplastin clotting time). Lupus anticoagulant, which is linked with thrombotic tendency, was seen in 34 of these 44 patients [42]. Polyphosphates activate platelets and factor XII during systemic inflammation, and this ultimately leads to coagulation. Pathogen-induced mechanisms and elements of NETs (neutrophil extracellular traps) are also possibly involved in coagulation activation [38]. The initiation of endothelial inflammation throughout the systemic inflammatory response possibly enhances microthrombi formation as well [43].

Hypoxemia has procoagulant outcomes, and therefore it should also be contemplated as a possible cause of COVID-19 coagulation disorders. The diminished oxygen levels cause tissue factor transcription, resulting in vascular fibrin accumulation. Plasminogen activator inhibitor-1 is coactivated by hypoxia, and it acts by repressing fibrinolysis [44]. This has been well depicted, as per Figure 5 below. Additionally, Figure 6 shows a summary of the mechanisms involved in the development of cardiovascular complications in COVID-19.

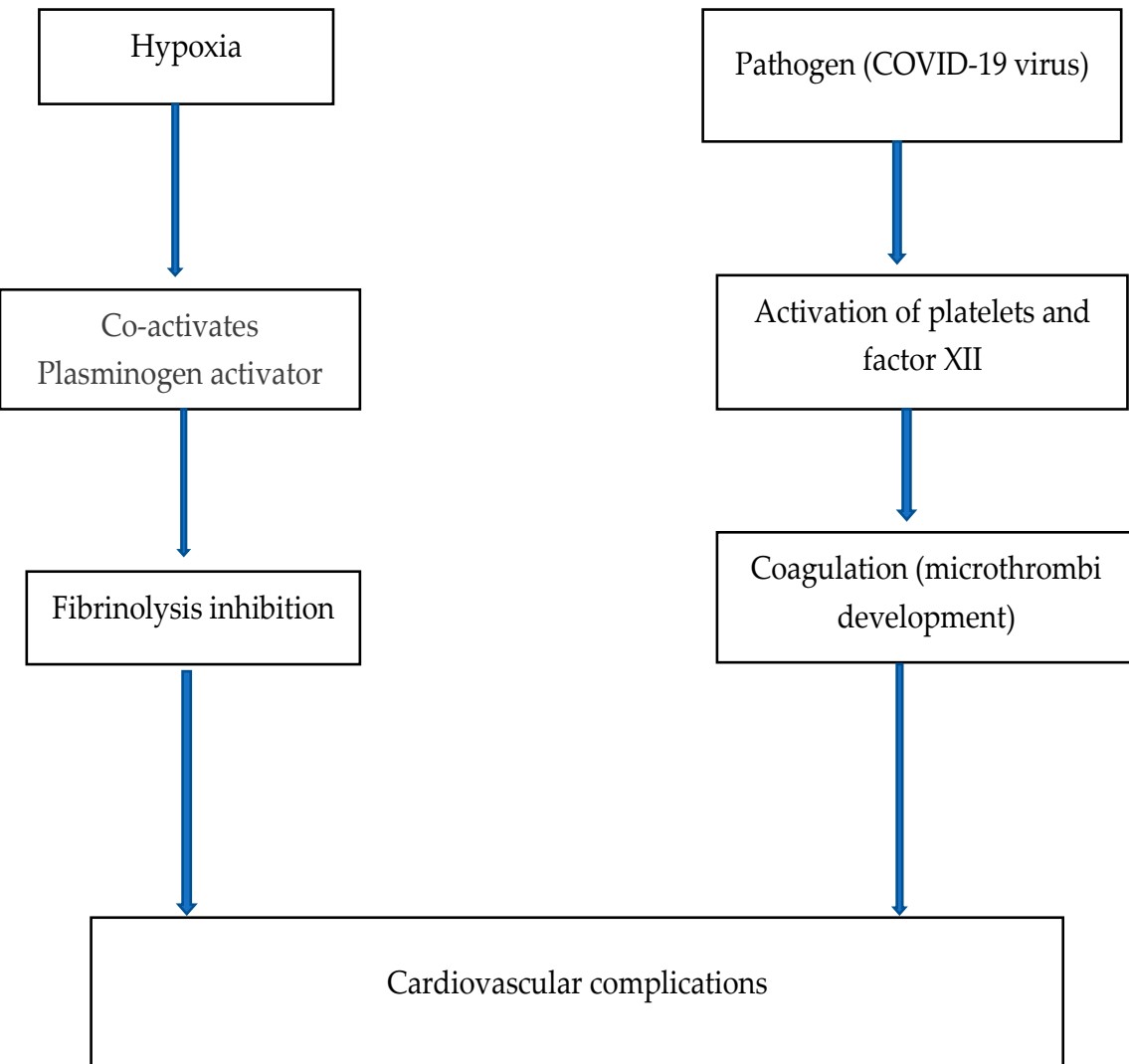

**Figure 5.** Diagrammatic illustration of prothrombotic state and hypercoagulability mechanism.

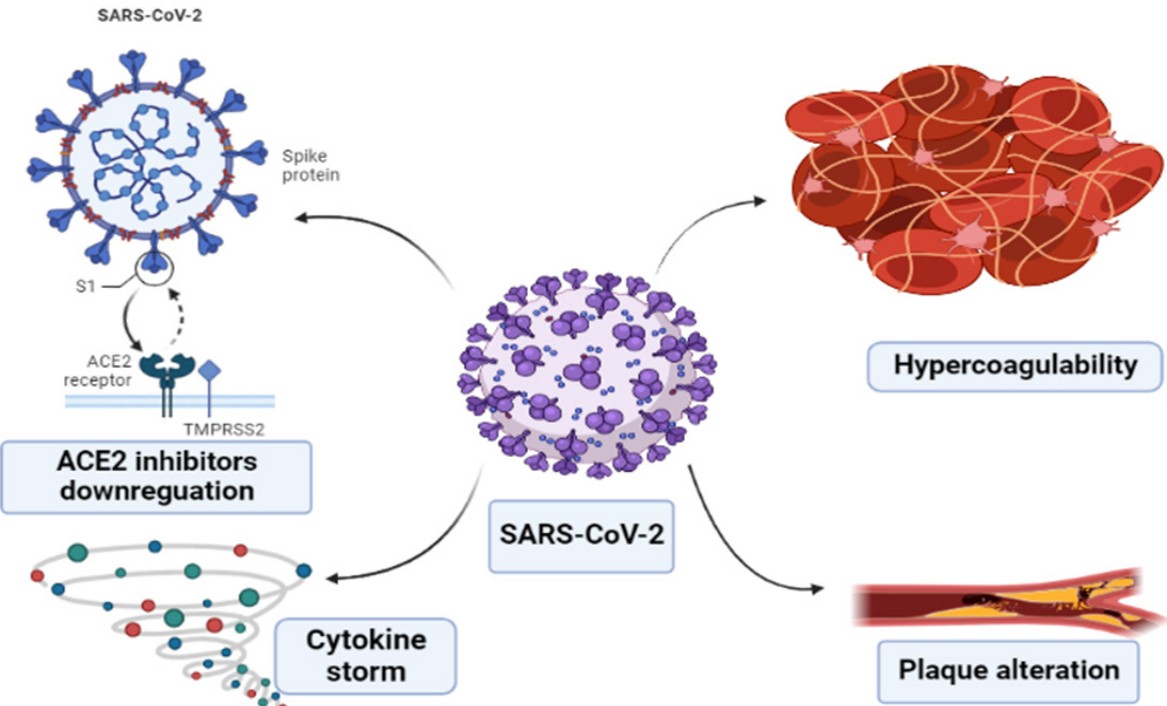

**Figure 6.** The possible mechanisms involved in the development of COVID-19 cardiovascular complications.

## 4. Cardiovascular Complications of COVID-19

### 4.1. Myocardial Infarction

MI (myocardial infarction) primarily occurs due to an obstruction of blood flow to the heart, resulting in the death of the heart tissues, and it is also referred to as heart failure [45]. Systemic inflammatory response syndrome enhances the possibility of plaque rupture, as well as thrombus development, which ensues in either non-ST-elevation MI or ST-elevation MI [46]. Various mechanisms have been proposed to be responsible for AMI in COVID-19 patients. A proinflammatory state may cause further coronary plaque damage, as was observed in the influenza outbreak, resulting in type-1 AMI [47]. An imbalance between the lowered oxygen supply and the elevated oxygen demand in the heart may be accountable for type-2 AMI [48]. A study on SARS-hospitalized patients revealed that acute MI was responsible for two out of five deaths [49]. Another study proved that STEMI patients admitted during the COVID-19 period had a higher rate of experiencing adverse events, as well as severe clinical manifestations [50]. Notably, there have been conflicting outcomes from the various pieces of research that have been conducted on STEMI patients' hospital admission during the COVID-19 pandemic. Those that reported a decline in AMI hospital admission during COVID-19 include [50,51], whereas there are some that reported an increase, such as [52,53]. Some of the likely explanations for this disparity include unequal global distribution admission rates, patient financial constraints, and economic instability in low GDP countries [54,55].

Early treatment and management delivery has been proposed to be an effective measure in preventing fatal outcomes in AMI patients with COVID-19 infection [56]. Table 1 below gives a summary of the available clinical evidence we found from our literature survey.

*4.2. Myocarditis*

Myocarditis is a condition where the heart muscles (myocardium) are inflamed, causing a myriad of clinical manifestations, such as chest pain, irregular heartbeats, and dyspnea. Notably, viral infections are mainly responsible for causing myocarditis in developed nations, whereas, in underdeveloped ones, both viruses and bacteria (diphtheriae) contribute to its pathogenesis [57]. The Dallas criteria (histopathological) and histological and immunological methods are currently used in diagnosis. The EMB diagnostic tool, which combines immunohistochemical and histopathological methods, is the best method in myocarditis diagnosis [58]. Interestingly, myocarditis was noted during the SARS-CoV and MERS-COV epidemics [59,60]. The first case reports of SARS-CoV-2 lacked myocarditis assessment, which probably had an effect on the understanding of this complication in COVID-19 patients. ECG modifications and anomalous changes in cardiac enzymes, which point to acute myocardial injury in COVID-19 patients, were observed in studies by Wang et al. and Doyen et al. [4,61]. The first myocarditis case was noted in a 63-year-old male who had no history of hypertension or any cardiac issue, and both his IL-6 and myocardial injury biomarkers (such as troponin I, NT-BNP) levels were elevated, as reported by Zeng et al. [62]. Subsequent cases linking COVID-19 to myocarditis in COVID-19 patients below the age of 40 include [63,64], and those above age 40 include [65,66]. Some autopsy reports also confirmed the presence of myocarditis in studies by Wichmann et al. and Rapkiewicz et al. [67,68]. However, another study revealed that the mortality rates in COVID-19 patients due to myocarditis were lower compared to other cardiac-related comorbidities [69]. The main mechanisms for myocarditis and myocardial injury are cardiomyocytes destruction due to viral entry, cytokine release syndrome, and hyperinflammation. Conflicting results have been observed in different pieces of research involving autopsy examinations of COVID-19 patients. Whether SARS-CoV-2 directly acts on cardiomyocytes or not is disputable; nevertheless, a direct effect on human stem cell-obtained cardiomyocytes has been proven [70]. This autopsy study analysis showed that the virus was not detected in the heart [71]. In contrast, a study involving 39 autopsy reports showed that there was myocardium viral presence in 24 cases [72]. Similarly, in another study, 1 out of the 23 cases showed viral RNA presence in the heart, along with lymphocytic myocarditis [73]. A report from England showed that eight children who had either been infected with COVID-19 previously had relatives who presented hyperinflammation, fever, and myocardial injury [74]. Other countries did report similar cases as well [75,76]. This prompted the naming of the condition as multisystem inflammatory syndrome in children (MIS-C).

*4.3. Arrhythmia*

Arrhythmia is among the early clinical manifestations of COVID-19, and it may suggest cardiac involvement once it is persistent. In a study by Liu et al. involving 137 patients, 7% of the patients had heart palpitations as a COVID-19 symptom [77]. A higher rate of arrhythmia (17%) was observed in another study Wang et al. involving positive COVID-19 cases [4]. Sinus tachycardia was the prevalent form of arrhythmia though the causal factor for this is yet to be known [78]. Atrial arrhythmias were recorded in 27.5% of COVID-19 ICU patients compared to zero in non-ICU patients [79]. Another study conducted by Hendren et al. on 700 COVID-19 patients showed that atrial fibrillation was linked with ICU admission [80]. Ventricular arrhythmias are also experienced in COVID-19 patients that are in a critical state [81]. The new development of tachyarrhythmia, followed by an increase in cardiac biomarkers in COVID-19 patients, may point to myocarditis [78,82].

### 4.4. Myocardial Interstitial Fibrosis

Elevated levels of myocardial fibrosis were observed in patients with COVID-19 as well as those who had recovered when magnetic resonance imaging of the myocardium was conducted. Furthermore, there seemed to be an association between these elevated levels and the severity of the COVID-19 disease [63,83]. Myocardial fibrosis may be influenced by various cardiac issues (such as heart failure and heart attack), but its classic development marker is enhanced fibroblast activation, which is due to higher measures of transforming growth factor β1 (TGF-β1) [84].

Interleukin-1 receptor-like 1 (also referred to as ST2 or IL1-RL1) is superficially located on the fibroblast cell, as well as on cardiomyocytes, and it binds to Interleukin-33 (IL-33). TGF-β1 activates IL1-RL1 into its dissolved form, which, on binding with IL-33, prevents the expected cardioprotective effect [85–87]. Myocardial fibrosis was also found to be a significant risk factor in determining the severity of cardiovascular complications in COVID-19 patients [88].

CMR detected diffuse fibrosis in a COVID-19-infected 45-year-old female with no myocarditis history, and palpitations were among the symptoms experienced 3 months post-COVID-19 infection [89]. This was confirmed by another report from a 49-year-old male who experienced dyspnea 6 weeks after the onset of COVID-19 clinical manifestation [90]. Furthermore, the autopsy results of 14 COVID-19 patients showed that six patients had focal cardiac fibrosis, although all had a history of MI [91]. However, some reports have been contradictory, such as in a case where edema was observed in two patients that possibly experienced myocarditis as a result of COVID-19, and the fibrosis was undetected [71].

### 4.5. Endothelial Cell Dysfunction and Vasculitis

Endothelial cells (ECs) are significant in the various pathologies of COVID-19 because they are involved in coagulation, immune response, and platelet functioning [92,93]. The existence of SARS-CoV-2 in the endothelial cells of various organs was seen in the post-mortem examinations of COVID-19 patients [94]. Viral infections of the ECs can result in endothelial damage and the destruction of vascular structure, ultimately causing internal leakage. Hyperinflammation, together with hypercoagulability, was seen in COVID-19 patients whose ECs were dysfunctional [20,95].

Elevated ACE-2 expression was seen in the postmortems of infected patients, along with cell distension, disrupted endothelial structure, and the disorientation of cells from the membrane, which leads to platelet aggregation [96]. scRNA-Seq analysis has shown that the genes linked with cytokine production, leukocyte induction, and immunomodulation are seen in ECs, and notably, they have a greater expression in the lungs than in other organs, which may insinuate that ECs are associated with inflammation to an extent [97]. Structural interference with ECs causes the platelets to bind with the membrane, leading to thrombosis, which is also seen in COVID-19-infected patients [98]. Oxidative stress on the ECs may lead to COVID-19-induced cardiovascular complications. When this occurs along with inflammation, thrombus formation becomes the resultant effect [99].

### 4.6. Thromboembolism

Thrombotic occurrences were noted to be frequent, particularly in ICU-admitted COVID-19 patients. A 40% rate of thrombotic event occurrence was reported in this study, with venous thromboembolism being higher than arterial thromboembolism [100]. Other studies confirmed this reality of thrombotic events (TEs) in critically ill COVID-19 patients [101,102]. A postmortem examination of four COVID-19 patients showed the existence of sizeable emboli in the lungs, and other organs, such as the brain, had microthrombi [103]. Conspicuously, the existence of emboli has also been linked with increased severity of the condition and mortality as well [104–107]. TE aetiology is thought to be multifactorial, and this includes EC destruction, which causes collagen to be released, leading to platelet activation and, ultimately, thrombus formation. Additionally, hyperinflammation plays a role in mediating the onset of coagulopathy, as discussed

earlier [95]. Other causes that may lead to a hypercoagulable state include hyperferritinemia, which is linked with macrophage activation syndrome and dispersed intravascular coagulation [39,108].

### 4.7. Dysautonomia

This is a medical state that is a result of a dysfunctional autonomic nervous system (ANS), resulting in the collapse of the parasympathetic, as well as sympathetic, components of the ANS, and its clinical manifestations include orthostatic intolerance, chest pain, and dizziness among others. It has been noted in COVID-19 patients, and it possibly occurs due to the progression of the infection or its prolongation ('long' COVID) [109–111]. Postural orthostatic hypotension (POTS), which is characterized by symptoms like fatigue, chest pain, and even orthostatic intolerance, may be experienced by some COVID-19 patients, and this is due to autonomic dysfunction [112–114]. Autoimmunity and hypovolemia, among others, could be the reason for POTS in COVID-19, although this is yet to be sufficiently verified [96,112].

**Table 1.** Summary of cardiovascular complications noted in COVID-19 patients.

| Cardiovascular Complication | Available Clinical Evidence | Cause(s) |
|---|---|---|
| Myocardial infarction | Two out of five SARS deaths are due to acute myocardial infarction (AMI) [49]. | Type 1 AMI- Inflammation Type 2 AMI- Imbalance between the lowered oxygen supply and the elevated oxygen demand in the heart. |
| Myocarditis | Elevated IL-6 and myocardial injury biomarkers (such as troponin I and NT-BNP) levels were noted [64]. | Viral and bacterial infection. |
| Arrhythmia | 7–17% COVID-19 patients experienced arrhythmia [4,47]. | Myocardial inflammation. |
| Myocardial interstitial fibrosis | Elevated levels of myocardial fibrosis were detected, and it is used as a determinant of severity of cardiovascular complication in COVID-19 patients [63,83]. | Enhanced fibroblast activation due to higher measures of transforming growth factor β1 (TGF-β1) [84]. |
| Endothelial cell dysfunction and Vasculitis | Disrupted endothelial structure was seen in the postmortems of infected COVID-19 patients [94]. | Oxidative stress on the endothelial cells. |
| Thromboembolism | A study showed a 40% occurrence of thrombotic events [100]. Postmortems of four COVID-19 patients showed the existence of sizeable emboli in the lungs, brain, and other organs [103]. | Platelet activation, hyperinflammation. |
| Dysautonomia | Noted in COVID-19 patients [109–111]. | Prolongation ('long' COVID) infection. |

## 5. Treatment

Table 2 below shows the various treatments that have been used in the treatment of COVID-19-induced cardiovascular complications with their outcomes, as seen from different case studies and reports.

**Table 2.** Treatment consideration table.

| Disease | Treatment | Outcomes |
| --- | --- | --- |
| Myocardial infarction | STEMI—(fibrinolytic therapy along with pharmacoinvasive strategy) [115] NSTEMI—aspirin, clopidogrel, enoxaparin, noradrenaline [116]. | Fibrinolytic therapy terminates MI Patient showed good therapeutic response within 2 days and was finally discharged after 22 days though some complications arose during the treatment. |
| Myocarditis | Lopinavir/ritonavir/hydroxychloroquine [117] IV prednisolone, and dexamethasone [118]. | ECG normalized, LVEF returned to 65%, patient discharged after 13 days. Prolonged duration before detection of viral RNA. |
| Myocardial interstitial fibrosis | Continuous positive airway pressure (CPAP) with a possibility of pacemaker implantation [119]. | N/A |
| ECs dysfunction and vasculitis | N-Acetyl-L-cysteine, Human recombinant IL-37 (antioxidant treatment) targeting IL-6/IL-6R [120]. | Lowered ROS formation Lowered incidents of pathogenesis of COVID-19. |
| Thrombotic events | Anticoagulation therapy + enoxaparin [121] Anti-coagulant therapy+ heparin [121]. | Patient was relieved of the abdominal pain and other symptoms There was improvement in the health of all the patients except one that died. |
| Arrhythmias | Antiarrhythmic agents (amiodarone) + procainamide or lidocaine If the patient is critical, defibrillation should be considered [122]. | Improved patient health. |
| Dysautonomia | Heart rate inhibitors (ivabradine) Sympatholytic drugs clonidine and methyldopa) Volume expanders (fludrocortisone and intravenous saline) [123]. | Quick betterment of the symptoms and lowered heart rate. |

*5.1. Anticoagulant Therapy*

Prophylactic anticoagulation is essential in the TE management of admitted COVID-19 patients. There has been a surge in both arterial thromboembolism- and venous thromboembolism (VTE)-hospitalized cases during the COVID-19 infection [102]. A study suggested that the VTE rate in hospitalized patients with COVID-19 may be higher than is contemplated [124]. However, the perfect thromboprophylaxis treatment is yet to be chosen. Figure 2 below has a summary of the currently available treatments.

Heparin was the earliest true anticoagulant to be discovered and it has been used in the treatment of conditions such as PE and DVT; however, some of its major drawbacks include bleeding, hyperkalaemia, and thrombocytopaenia [125].

A study involving 449 patients with acute COVID-19 revealed no changes in the 28-day mortality of 99 patients after low molecular weight heparin had been administered for a minimum of seven days, although changes were observed in the subgroups. The heparin subgroup experienced lesser mortality in comparison to the non-heparin group [126].

Enoxaparin was endorsed as the drug choice to be used in acute-phase VTE, whereas rivaroxaban and apixaban are to be chosen in postdischarge prophylaxis in those COVID-19 patients at a greater risk of developing thromboembolic events [127].

*5.2. ACE Inhibitors and Angiotensin Receptor Blockers*

SARS-CoV-2 binds to ACE-2 receptor prior to its entrance to the cell [20], and using the renin-angiotensin-aldosterone system (RAAS) inhibitors, which may raise ACE-2 expression, has raised doubts among researchers [128].

It is yet to be discovered if the greater expression of ACE-2 enhances predisposition to SARS-CoV-2 infection, but it was suggested that the use of RAAS inhibitors in COVID-19-stable patients should be continued [129].

The presence of divergent outcomes in both the preclinical and clinical studies that evaluated the effect of RAAS inhibition on ACE-2 expression has increased speculation [130]; however, this was clarified by a study that was carried out in the COVID-19 context. According to this study, it was made clear that various RAAS inhibitors have distinct effects on ACE-2 expression, hence the varying outcomes [131]. In a study conducted on mouse models, the decreased expression of ACE-2 showed a greater extent of lung damage caused by influenza [131]. Furthermore, in relation to COVID-19, a comprehensive study showed that the administration of RAS inhibitors in outpatients in no way enhanced the chances of COVID-19 diagnosis, its severity, or even the probability of hospitalization [132].

Angiotensin-receptor blockers and mineralocorticoid-receptor blockers enhance ACE-2 expression and activity both in preclinical and clinical studies [133]. However, ACE inhibitors only enhance expression but have no effect on activity, as seen in experimental models [130,131].

### 5.3. Immunosuppressive Therapy

The regular use of corticosteroids is not endorsed in SARS-CoV-2, as was the case in the SARS and MERS epidemics. This is because they may aggravate the already occurring lung injury [134]. Nonetheless, the injurious effects of the cytokine storm in COVID-19-infected patients have prompted the use of immunosuppression to control the hyperinflammatory response. Screening the infected patients and evaluating their laboratory values (such as platelet count and erythrocyte sedimentation rate) and also applying the use of hemophagocytic response scores, which are used in hyperinflammation diagnosis, may be beneficial in recognizing patients that need immunosuppression for an improved outcome [135,136].

IL-6 inhibitors, such as tocilizumab, may be useful as immunomodulators. In a case series involving 21 COVID-19 cases, tocilizumab ameliorated the symptoms, and it did prevent further worsening of the patients [137].

Furthermore, according to a study by Thakur et al., it was observed that the use of steroids among hospitalized patients proved to be effective as the number of COVID-19 deaths was reduced [138].

### 5.4. Mechanical Support

It was noted that some COVID-19 patients developed shock [4,139]. This prompted the need for mechanical circulatory support, and, when necessary, pulmonary support should be included [140]. Some of the techniques that should be used include left ventricular assist devices (LVADs) and venoarterial extracorporeal membrane oxygenation, which are very important, particularly in critical conditions of respiratory failure [141,142]. In the 2009 influenza epidemic, 13 myocarditis patients were assessed, and nine out of them, who were using an intra-aortic balloon pump (IABP), which is a mechanical circulatory support, survived, whereas the remaining four that were not on mechanical circulatory support succumbed [143]. A case has also been reported of a COVID-19 patient experiencing cardiogenic shock who recovered after being on IABP support [144]. The selection of patients who are to receive the various mechanical support systems should consider any other comorbidities being experienced by the patient in order to help prevent aggravating the patient's condition.

### 5.5. Antiviral Drugs

Most of the drugs being used in the treatment of COVID-19 are similar to those used in the SARS and MERS epidemics. However, some new antiviral drugs, like chloroquine, have also been embedded in treatment to ensure enhanced, effective treatment. A blend of two or more antiviral drugs is not advocated for when considering toxicity, which may be incurred. There is still a lack of sufficient quality clinical evidence about this therapy, although a few studies have depicted a desirable outcome in COVID-19 patients having cardiovascular complications. An example of the antiviral drugs currently being used is briefly discussed below and shown in Figure 7.

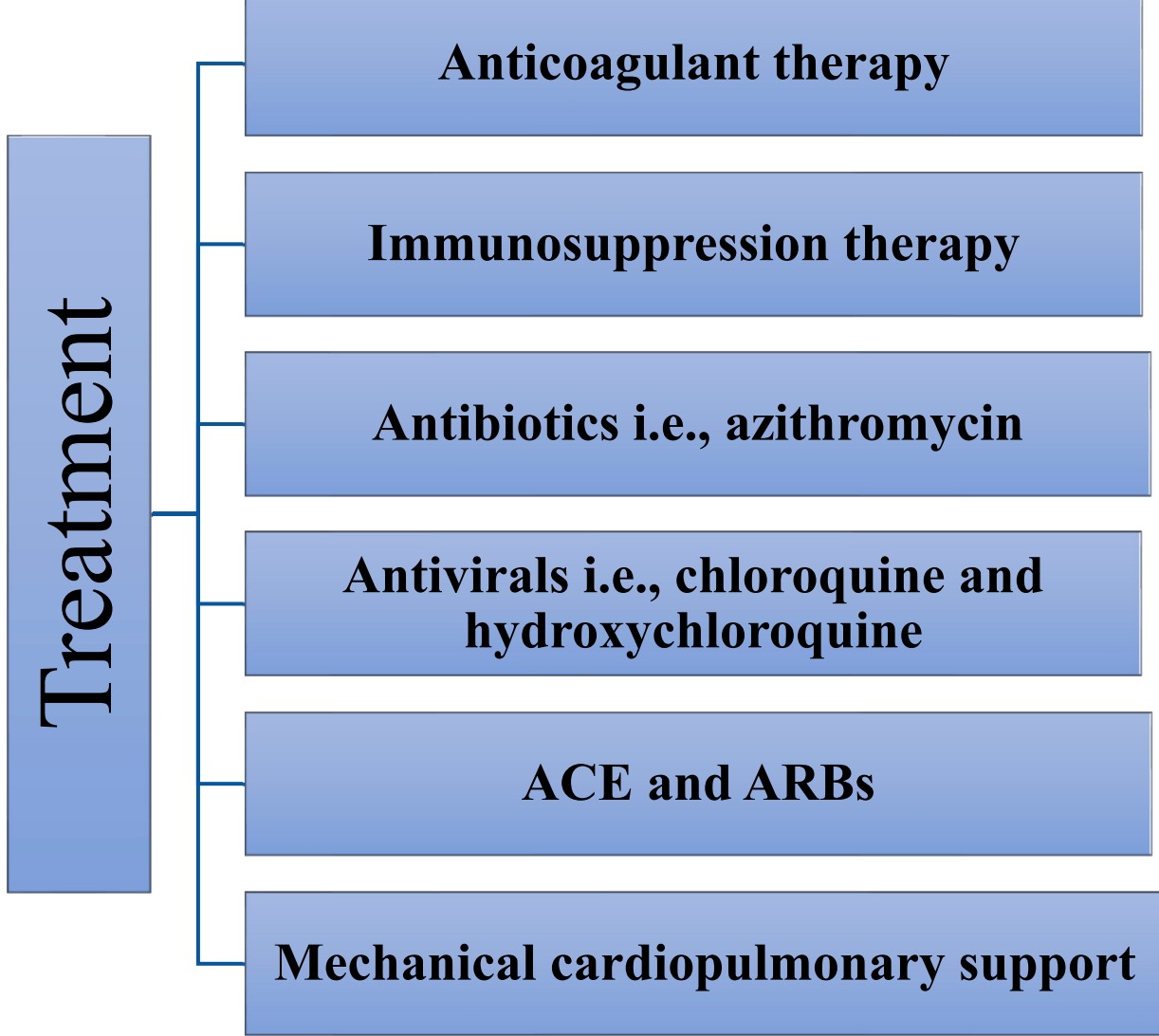

**Figure 7.** Currently available treatments.

Chloroquine and Hydroxychloroquine

These are basically meant for malarial treatment, and they are of the quinoline family. Additionally, they can be used in lupus erythematosus, as well as in rheumatoid arthritis treatment [145]. Chloroquine showed a positive outcome as a potential drug to be used in SARS, and recently, another study revealed that chloroquine could adequately impede SARS-CoV-2 by restraining ACE2 glycosylation by elevating the endosomal pH [146]. The Italian treatment guidelines approved the use of this combination, but the FDA, which had, earlier on, accepted this combination, revoked its emergency use after clinical trials had been conducted [147,148].

*5.6. Antibiotics*

Azithromycin

This is an antibiotic that inhibits inflammatory processes [149]. It was effective in both Zika virus and Ebola virus treatment [150]. It has been suggested that it should be used together with hydroxychloroquine, although the mechanism is not well established yet [151].

Chloroquine and azithromycin showed a positive outcome; however, their main adverse effect is QTc prolongation, which may be fatal, especially in COVID-19 patients. However, guidelines have been created with respect to handling Qt prolongation in COVID-19 patients [152].

### 6. Conclusions

The development of cardiovascular complications in COVID-19 patients worsened the pandemic situation. The lack of oxygen cylinders and the exacerbation of the symptoms, particularly in less developed nations, proved fatal. Therefore, this review was mainly meant to elucidate the proposed mechanisms in the pathogenesis of cardiovascular complications in COVID-19 patients, as well as the possible effective treatment modalities that have already been used and have had a notable positive therapeutic effect. Future studies should include larger cohort studies with prompt reporting of patients' responses to the drugs. This will ultimately help better our comprehension of the complications and the effective treatments. Finally, there should be an increase in preclinical studies for better validifying the clinical results obtained.

We recommend that the knowledge and aptitude of the physicians be strengthened regularly through seminars, news, etc., on the latest advancements. Additionally, the judicious practice of medical ethics should be considered, especially in special conditions, such as comorbidities, terminally ill patients, etc.

A possible limitation of this study is that we did not use the traditional system of review, where all evidence is counted to be equal. On the other hand, our work favored research that has already had a high impact. Therefore, it is possible that we did not add some other references that are true but have less impact.

**Author Contributions:** I.O.: Conceptualization, Methodology, Writing—original draft preparation. M.V.: Data curation, Figure(s) preparation. M.A.: Visualization, Investigation. A.S.: Supervision, Review and Editing; P.K.: Validation. All authors have read and agreed to the published version of the manuscript.

**Funding:** This review received no external funding.

**Institutional Review Board Statement:** Not applicable.

**Informed Consent Statement:** Not applicable.

**Data Availability Statement:** Not applicable.

**Acknowledgments:** We thank ISF College of Pharmacy, Moga for providing the necessary facilities required during this review.

**Conflicts of Interest:** The authors declare no conflict of interest.

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
