# Peer review of "A Scoping Review on COVID-19-Induced Cardiovascular Complications"

_covid, doi:10.3390/covid3030026_

Round 1
Reviewer 1 Report
The topic you have chosen is certainly of considerable interest, but clearly needs to be better expressed.
Sections 3 and 4 are the ones of greatest interest for your scoping review, and also the least clear and most confusing.
You must try to set out your considerations of these two sections better, perhaps with the help of some diagrams or tables, as you did so well in section 5.
In fact, section 5, the one in which you set out the treatment is very schematic and well summarised.
In the final part of the conclusions, you talk in general about covid therapies, which are clearly not the purpose of your discussion. Please try to be more concise.
The article needs to be better organised.
Author Response
Response to Reviewer 1 Comments
Point 1: Sections 3 and 4 are the ones of greatest interest for your scoping review, and also the least clear and most confusing.
You must try to set out your considerations of these two sections better, perhaps with the help of some diagrams or tables, as you did so well in section 5.
Response 1: We are thankful for this suggestion.
For section 3, four new figures have been added at end of each sub-section. The figures are concise and they give an overview of the explanations in the sub-sections.
For section 4, we have provided a summary of our findings in table form. The table is summarized and it is easy to read and understand.
Point 2: In the final part of the conclusions, you talk in general about covid therapies, which are clearly not the purpose of your discussion. Please try to be more concise.
Response 2: Thank you once again for the wonderful comment. We have made the adjustment and have tried to be more concise. We have deleted the general information that had been written.
Reviewer 2 Report
The authors reported the available information on COVID-19 induced cardio-vascular complications. The manuscript is detailed, the authors included 44 studies in the final analysis. The mechanisms of cardiovascular complications as well as the treatment possibilities are reviewed.
There are a few sessions, which are not completely clear and needs to be clarified. For example, more information about how SARS-CoV-2 regulates ACE-2 receptors would help the understanding of chapter 3.1. In this session, rows 74-78 are not clear.
Figure legends and numbers should be reviewed. In rows 88-90 the authrs refer to Figure 1, which should be about the ACE-2 expression of cardiomyocytes, endothelial cells and pericytes, but I did not fid such a figure and in row 60 the authors refer to the study design as Figure 1. This does not have figure legend, so for this reason I am not sure, if the study design is the Figure 1. In row 151 the authors mention aPTT, but it is not clear from the manuscript, what is this stand for. Some sentences need to be reviewed and changed for better understanding (e.g.217-219).
These are only examples, but in general the manuscript needs to be reviewed and edited before publication, therefore I suggest to accept the manuscript after minor revision.
Author Response
Response to Reviewer 2 Comments
Point 1:
There are a few sessions, which are not completely clear and needs to be clarified. For example, more information about how SARS-CoV-2 regulates ACE-2 receptors would help the understanding of chapter 3.1. In this session, rows 74-78 are not clear.
Response 1: thank you for the wonderful comment. We have made the adjustment by adding figures which are in summarized form and easy to understand.
Point 2: Figure legends and numbers should be reviewed. In rows 88-90 the authrs refer to Figure 1, which should be about the ACE-2 expression of cardiomyocytes, endothelial cells and pericytes, but I did not fid such a figure and in row 60 the authors refer to the study design as Figure 1. This does not have figure legend, so for this reason I am not sure, if the study design is the Figure 1. In row 151 the authors mention aPTT, but it is not clear from the manuscript, what is this stand for. Some sentences need to be reviewed and changed for better understanding (e.g.217-219).
Response 2: We highly appreciate this valuable comment and the changes have been made as requested.
Figure legends have been adjusted as required along with the corrections made.
Sentences 217-219 have been adjusted accordingly. The full form for aPTT time (Activated Partial Thromboplastin Clotting Time) has been provided.
Round 2
Reviewer 1 Report
I appreciated the changes made, it is definitely more accessible now, but some concepts still need to be clarified.
In section 3.1 you are saying that covid decreases the expression of ACE-2 receptors and since as a result there is an imbalance with the circulating ACE enzyme there will be cardiovascular consequences? In this case you need to express this better by adding it to the paragraph, which is not clear at all this way.
In Figure 5, it appears that hypoxia promotes fibrinolysis, whereas in line 216 you have stated the opposite, so correct.
In Table 1, it is not clear what sense these remarks make “Viral infection- mostly affects developed nations, Viral and bacterial infection- mostly affects underdeveloped nations”, so either explain it or replace or remove them.
Same thing for table 2, what does it mean “Myocardial interstitial fibrosis Continuous positive airway pressure (CPAP) with a possibility of pacemaker implantation [119]”?
Should patients with interstitial myocardial fibrosis be submitted to NIV? Either explain it better or replace or remove them.
Author Response
Response to Reviewer 1 Comments
Point 1: In section 3.1 you are saying that covid decreases the expression of ACE-2 receptors and since as a result there is an imbalance with the circulating ACE enzyme there will be cardiovascular consequences? In this case you need to express this better by adding it to the paragraph, which is not clear at all this way.
Response 1: We are thankful for this suggestion.
Covid decreases the expression of ACE-2 receptors and this increases angiotensin II levels. The imbalance between ACE-2 and angiotensin II levels leads to the development of cardiovascular complications. I have made a few modifications as well on the manuscript and I hope the information is clearly communicated.
Point 2: In Figure 5, it appears that hypoxia promotes fibrinolysis, whereas in line 216 you have stated the opposite, so correct.
Response 2: Thank you once again for the wonderful comment. I omitted the word inhibition but I have added it now in the new figure provided.
Point 3: In Table 1, it is not clear what sense these remarks make “Viral infection- mostly affects developed nations, Viral and bacterial infection- mostly affects underdeveloped nations”, so either explain it or replace or remove them.
Response 3: Thank you for the suggestion. I have deleted the information on nations. The information I wanted to relay was that viral infections mostly affect people living in developed nations whereas both viral and bacterial infections affect people living in underdeveloped nations.
Point 4: Same thing for table 2, what does it mean “Myocardial interstitial fibrosis Continuous positive airway pressure (CPAP) with a possibility of pacemaker implantation [119]”?
Should patients with interstitial myocardial fibrosis be submitted to NIV? Either explain it better or replace or remove them.
Response 4:
Thank you for this comment. CPAP is a modality of NIV treatment and according to the cited article, the possibility of using a pacemaker should be assessed based on the level of severity and response of the patient during treatment. I hope I have relayed this information in the simplest form in the manuscript and in case it is still confusing then I will consider removing it upon your request.
I highly appreciate the amazing comments and your speedy response in reviewing the manuscript. Thank you.